# The United Kingdom Research study into Ethnicity And COVID-19 outcomes in Healthcare workers (UK-REACH): protocol for a prospective longitudinal cohort study of healthcare and ancillary workers in UK healthcare settings

Katherine Woolf  ,[1] Carl Melbourne,[2] Luke Bryant,[3] Anna L Guyatt,[2] I Chris McManus  ,[1] Amit Gupta,[4] Robert C Free,[3,5] Laura Nellums  ,[6] Sue Carr,[7,8] Catherine John,[9] Christopher A Martin  ,[3,10] Louise V Wain,[5,9] Laura J Gray,[9] Claire Garwood,[3] Vishant Modhwadia,[3] Keith R Abrams,[11] Martin D Tobin,[2,12] Kamlesh Khunti,[13] Manish Pareek,[3,10] on behalf of the UK-REACH Study Collaborative Group+

KW, CM and LB are joint first authors.

For numbered affiliations see end of article.

**Correspondence to**
Dr Manish Pareek;
manish.pareek@leicester.ac.uk

## ABSTRACT

**Introduction** The COVID-19 pandemic has resulted in significant morbidity and mortality and devastated economies globally. Among groups at increased risk are healthcare workers (HCWs) and ethnic minority groups. Emerging evidence suggests that HCWs from ethnic minority groups are at increased risk of adverse COVID-19-related outcomes. To date, there has been no large-scale analysis of these risks in UK HCWs or ancillary workers in healthcare settings, stratified by ethnicity or occupation, and adjusted for confounders. This paper reports the protocol for a prospective longitudinal questionnaire study of UK HCWs, as part of the UK-REACH programme (The United Kingdom Research study into Ethnicity And COVID-19 outcomes in Healthcare workers).

**Methods and analysis** A baseline questionnaire will be administered to a national cohort of UK HCWs and ancillary workers in healthcare settings, and those registered with UK healthcare regulators, with follow-up questionnaires administered at 4 and 8 months. With consent, questionnaire data will be linked to health records with 25-year follow-up. Univariate associations between ethnicity and clinical COVID-19 outcomes, physical and mental health, and key confounders/explanatory variables will be tested. Multivariable analyses will test for associations between ethnicity and key outcomes adjusted for the confounder/explanatory variables. We will model changes over time by ethnic group, facilitating understanding of absolute and relative risks in different ethnic groups, and generalisability of findings.

**Ethics and dissemination** The study is approved by Health Research Authority (reference 20/HRA/4718), and carries minimal risk. We aim to manage the small risk of participant distress about questions on sensitive topics by clearly participant information that the questionnaire covers sensitive topics and there is no obligation to answer these or any other questions, and by providing support organisation links. Results will be disseminated with reports to Government and papers submitted to pre-print servers and peer reviewed journals.

**Trial registration number** ISRCTN11811602; Pre-results.

### Strengths and limitations of this study

► Sampling frame includes a variety of healthcare worker job roles including ancillary workers in healthcare settings will improve the generalisability of results across diverse healthcare job roles.

► Longitudinal study including three waves of questionnaire data collection, and linkage to administrative data over 25 years, with consent, will enable researchers to infer causal relationships.

► Unique support from all major UK healthcare worker regulators, relevant healthcare worker organisations and a Professional Expert Panel to increase participant uptake and the validity of findings.

► Potential for self-selection bias and low response rates.

► The use of electronic invitations and online data collection makes it harder to reach ancillary workers without regular access to work email addresses.

## INTRODUCTION

COVID-19 has spread rapidly across the world, causing significant morbidity and mortality and devastating health economies in many countries. Healthcare workers (HCWs) have been identified as being at increased risk of SARS-CoV-2 infection and adverse outcomes,[1–3] as have individuals from

ethnic minority groups.[2 4–13] Emerging evidence suggests that ethnic minority groups may also be at greater risk of long-term COVID-19 sequelae.[4] HCWs and individuals from ethnic minority groups may also be at increased risk of COVID-19-related poor mental health outcomes, including anxiety, depression and post-traumatic stress.[14–19]

There are concerns that HCWs from ethnic minority groups are at particular risk of SARS-CoV-2 (severe acute respiratory syndrome coronavirus 2) infection and adverse COVID-19 outcomes compared with white HCWs[3 20 21] However, the quality of data relating to COVID-19 outcomes in HCWs remains poor, with no large representative studies of the risk of COVID-19 infection or adverse outcomes in healthcare workers or ancillary workers in healthcare settings (hereafter 'HCWs') stratified by ethnicity or occupation type, controlling for potential confounders.

To address this, UK-REACH (United Kingdom Research study into COVID-19 outcomes in Healthcare workers) will rapidly examine differences in COVID-19 diagnosis, clinical outcomes (diagnosis, hospitalisation, ICU admission), professional roles and well-being among ethnic minority and white HCWs through a unique large database analysis (rapid linkage and analysis of anonymised professional registration and National Health Service (NHS) datasets), longitudinal cohort study, legal/ethical analysis and qualitative work packages. This work will provide information on very short-term outcomes and produce rapid actionable outputs as well as enabling investigations of the medium/long-term effects of COVID-19 on HCWs in future studies through the linkage and cohort study. This protocol describes the UK-REACH longitudinal cohort study.

## Research question

What is the impact of COVID-19 on the physical and mental health of ethnic minority HCWs compared with white HCWs in the short term and the longer term?

## Aims

To examine the relationship between ethnicity and COVID-19-related mental and physical health outcomes through the establishment of a novel longitudinal cohort study of HCWs, including recruitment from the membership bodies and professional registers for different healthcare roles, providers of facilities management, and directly from UK healthcare settings.

To study changes in health outcomes, social circumstances and professional roles of HCWs of different ethnicities, over the course of the COVID-19 pandemic and to characterise longer-term outcomes and consequences.

To measure differences in the impact of COVID-19 infection and working during the pandemic on physical and mental health in a multi-ethnic group of HCWs in the UK.

## Objectives

► To survey HCWs at baseline to collect data on demographics, job role, attitudes to work and work climate, social and living circumstances, values and personality and physical and mental health.
► To collect baseline biological samples for future analysis in a subsample of consenting participants.
► To conduct follow-up surveys and samples over 12 months in order to capture changes over subsequent COVID-19 pandemic waves.
► To link survey data to participant healthcare records, with consent.

## METHODS AND ANALYSIS

### Study design

National prospective longitudinal cohort study in all four nations of the UK.

### Setting

HCWs and ancillary workers in healthcare settings within the UK.

### Participants

#### Inclusion criteria

Age≥16 years.

Living in the UK.

HCW or ancillary worker in a UK healthcare setting OR Registered with the following UK healthcare professional regulatory bodies: the General Medical Council (GMC), Nursing and Midwifery Council (NMC), General Dental Council (GDC), Health and Care Professions Council (HCPC), General Optical Council (GOC), General Pharmaceutical Council (GPC) or the Pharmaceutical Society of Northern Ireland (PSNI).

Willing and able to give informed consent.

#### Exclusion criteria

Age <16 years.

Living outside the UK.

Not a HCW or ancillary worker in a healthcare setting AND not regulated by one of the professional regulatory bodies listed above.

Unwilling and/or unable to give informed consent.

#### Sample size

We aim to recruit at least 32,000 HCWs (66% from ethnic minority groups). See figure 1 for the study flow chart. The proposed sample will approximately comprise:
► 10 000 doctors.
► 10 000 nurses, midwives and nursing associates.
► 4000 ancillary workers.
► 2000 allied health professionals.
► 2000 ambulance workers.
► 2000 pharmacists and pharmacy technicians.
► 1000 dentists and dental care professionals.
► 1000 optometrists and dispensing opticians.

While the above numbers represent our target numbers for recruitment, we will welcome participants working in other roles within healthcare settings. We may adapt

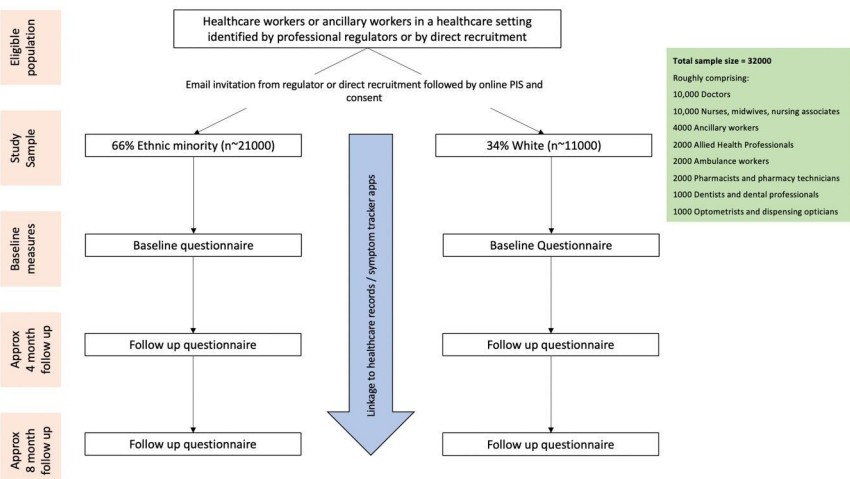

**Figure 1** Study flowchart. PIS, participant information sheet.

sampling frame for subgroups following initial response, to ensure we have appropriate representation of ethnic minority participants in each job role.

This sample size has been pragmatically chosen to allow for adequate representation of all ethnic groups within all job-role categories. Based on general population prevalence estimates[11] producing conservative power estimates, we anticipate at least 80% (statistical) power to detect modest effects of ethnicity (Relative Risk (RR) ≥1.5) for more prevalent outcomes (eg, COVID-19 diagnosis) and larger effects (RR ≥2) for rarer outcomes (eg, mortality). Power calculations will be reviewed to reflect changing rates of infection.

### Recruitment

Recruitment will be via several routes and will be incentivised by the inclusion of a prize draw for those who complete the questionnaire:

1. Email, letter and/or text message invitations from healthcare professional regulators.
   The records and registrations with the GMC, NMC, GDC, HCPC, GOC, GPC, PSNI will be used as a sampling frame. Where possible, we will use demographic data routinely collected by regulators to oversample for people identifying as being from ethnic minority backgrounds. We will endeavour to sample representatively across age groups, sex/gender, job roles and other characteristics, in order to maximise the generalisability of our findings. Regulators will send invitations and reminders on behalf of the principal investigator. Alongside these invites, regulators will also promote the study through their social media channels as the invites go out.

2. Targeted advertisement to key staff groups through healthcare organisations.
   We will advertise the study through the general communication channels of regulators, professional bodies (eg, Royal College of Midwives, Royal College of General Practitioners), Health Education bodies and other relevant organisations (eg, the British Medical Association, the British Association of Physicians of Indian Origin, The Filipino Nurses Association United Kingdom). This will include promotion through newsletters, web pages, email communications and banners on self-service portals (eg, on the NHS Electronic Staff Record portal used for accessing payslips).

3. General publicity of the study via print and broadcast media, social media for the study and other relevant organisations and posters or flyers in workplaces, as relevant to participant staff groups.

4. Direct invitation and recruitment via UK healthcare providers.
   UK healthcare providers, including at least 30 NHS Hospital Trusts, will advertise to potential participants by email, text, mail, verbally or through posters/flyers. Trusts will be selected to represent a range of geographical areas (to include England, Scotland, Wales and Northern Ireland). We will aim to recruit from both large teaching hospitals and smaller community healthcare trusts and will take into consideration regional ethnic diversity when selecting trusts. We will also use a study infographic to promote recruitment through NHS trusts. Recruitment of ancillary staff has been facilitated by Serco at specific trusts, through utilisation of content handouts and posters which have been cascaded by contractors on site.
   Invitations and advertisements will direct staff via a weblink and/or QR code to the study recruitment site. In the case of direct recruitment through UK healthcare settings, potential participants will be supported to join with the help of suitably trained members of the local research team where appropriate. Reminders will also be sent to improve recruitment to the study.

### Data collection

See figure 2 for the study timeline.

To gain informed consent, potentially interested participants will have the opportunity to read the UK-REACH participant information sheet (PIS) online via a web application, or in person with a member of the local

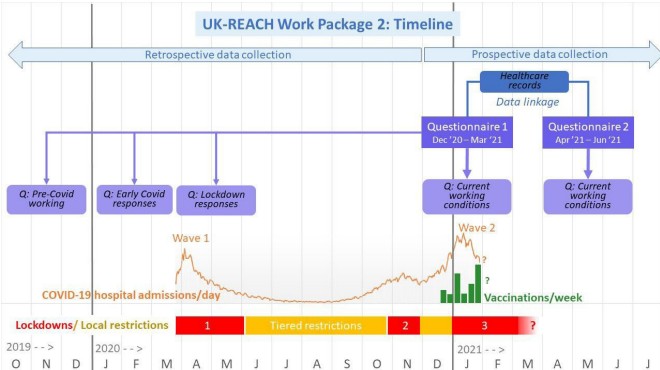

**Figure 2** UK-REACH Work package 2 timeline as of 4 February 2021. Dates are shown across the bottom from October 2019 to July 2021. The COVID-19 outbreak began in the UK in January 2020 with numbers of daily hospital admissions due to COVID-19 shown by the lower orange line for Wave one and Wave two. Vaccinations began in December 2020 and are shown by the green line for daily vaccinations. Lockdowns are shown by horizontal red bars, the first national lockdown beginning on 23 March 2020, the second on 5 November and the third of 5 January 2021. Lockdowns differed somewhat in timing between England, Wales, Scotland and Northern Ireland. Tiered local restrictions were in place in various regions of the UK between lockdowns, shown in yellow. Questionnaire 1 of Work package 2 began to be distributed on 4 December onwards and is being distributed until the end of March 2021. Questionnaire 1 asks about current events and working conditions, as well as retrospectively about events and working conditions pre-Covid in 2019, about early COVID-19 responses in the first months of 2021 and about events during the first national lockdown. Questionnaire 2 will be distributed 4 months after registration for questionnaire 1 and therefore will be distributed between April and June 2021. Questionnaire 2 asks primarily about current working conditions, and changes to other aspects of participants' lives captured in Questionnaire 1, including key measures of physical and mental health. With consent, the questionnaire data will be linked to electronic healthcare record data. UK-REACH, The United Kingdom Research study into Ethnicity And COVID-19 outcomes in Healthcare workers.

research team, before deciding if they would like to take part. Potential participants will be asked to register for a UK-REACH account (https://uk-reach.org/) by entering an email address and password, then will be asked to sign an online consent form. To complete registration, participants will be asked to provide personal details including name, date of birth and address. Participants will be considered enrolled into the study after completing the registration process and giving their electronic consent on a secure web page. They will be assigned a unique identifier at this point. This will be securely retained in an electronic database at the University of Leicester, which will also function as an enrolment log.

Once translation has been undertaken and checked, we will make study materials (PIS and consent form) available in alternative languages commonly used by workers

in healthcare settings as required. These will be made available on the UK-REACH website.

After consenting to participation in the study, participants will be asked to complete a baseline online questionnaire on demographics, job role and other work circumstances, location of work/residence, interaction with patients with COVID-19 (including access to personal protective equipment (PPE), social and living circumstances, discrimination and harassment, values and personality and physical and mental health (see Measures, below). This questionnaire will be accessible via the study website and support to complete the questionnaire will also be available from the study team. Some identifiable data (eg, name, DOB, address) will be collected during registration and/or the questionnaire to facilitate secure data linkage (see Data linkage, below).

Participants will be invited to complete the baseline questionnaire between 1 December 2020 and January 2021. Participants will have the option of completing two further questionnaires, one at approximately 4 months from baseline and one at approximately 8 months from baseline. Consent will be requested to follow-up participants for up to 25 years (subject to funding), and during this time serial questionnaire data will be collected, and periodic linkage with healthcare records will take place (see Data linkage, below).

### Data linkage

Participants will be asked to provide consent for the study to follow their health by extracting information from their past and future NHS healthcare records (including NHS number), any COVID-19 related records and from 'COVID-19 symptom study' websites or apps[22] if they use them.

Initially, questionnaire and personal data, for which consent has been given, will be electronically transferred to University of Leicester and stored separately on secure servers. A study ID for each participant together with identifiable data will be sent to NHS Wales Informatics Service in order to acquire the list of NHS numbers/ Community Health Index (CHI) number (for Scotland)/ Health and Care number (for Northern Ireland) in partnership with the relevant UK NHS data providers. Study ID and NHS/CHI/H&C number will then be used to link to healthcare records by the relevant data provider across the UK. Deidentified records will be sent directly to the Secure Anonymised Information Linkage databank (SAIL, https://saildatabank.com), retaining Study ID. Separately, study ID and corresponding questionnaire data will be sent from the University of Leicester to SAIL. These datasets will be linked within SAIL to provide the complete dataset. Interim analysis of unlinked questionnaire data will be performed at the University of Leicester using pseudonymised datasets. All linked data analysis will take place within SAIL.

## Questionnaire design

The questionnaires will be designed by the study team with input from the Professional Expert Panel (PEP)—see Patient and Public Involvement, below. Where possible, previously used and/or validated measures are being used. The study team will devise measures where none are available.

The baseline questionnaire will ask about participants' current experiences and attitudes as well as collecting some retrospective data about participants' experiences and attitudes at the start of the pandemic and/or pre-pandemic. Follow-up questionnaires will repeat outcome measures, and may include new items relevant to the progression of the pandemic. The data dictionary (https://www.uk-reach.org/data-dictionary) contains the source(s) for each question. The baseline questionnaire is included as an Appendix.

## Patient and public involvement

The UK-REACH team have worked closely with national and local organisations representing ethnic minority HCWs, who have been closely involved in identifying the research questions, deciding the study methodology, and are included either as members of the study delivery team or collaborators. They are also part of the Stakeholder group (see online supplemental information for list of organisations) that will meet monthly to monitor study progress and research outputs and provide advice to the research team on study delivery and progress. This group will also be central in disseminating the research findings. Alongside the high-level national organisation input into this stakeholder group, we will have representation from ethnic minority HCWs, including those who have contracted COVID-19 (mild to severe), to provide feedback on our work and how it should be disseminated. Members of our stakeholder/public engagement group will also sit on the Scientific Committee to ensure there is a seamless flow of information from the research team and the public engagement/stakeholder group. In addition we are working closely with the Centre for BME Health (Leicester, UK) to ensure that we are working to engage staff from a range of ethnic groups.

A PEP will provide feedback on UK-REACH study materials and sampling methodology such as surveys, questionnaires and interview and focus group topic guides. The PEP is made up of individuals who work within a healthcare setting from a range of ethnic backgrounds, occupational backgrounds and genders. Staff have unique insight related to their professions or ethnic groups and are, therefore, in a position to provide advice and lived experience related to certain aspects of the project. The aim is for the PEP to draw on their experiences when providing their comments to ensure research instruments are at optimum suitability for study participants. The PEP meets virtually on a bimonthly basis via Microsoft Teams. Study items/documents for discussion are circulated a week in advance of the PEP meeting and the group's Chair and Co-chair (PEP members who both volunteered to take on the role) moderate the meeting. UK-REACH team members are present in order to answer study-specific queries, and so only enter the discussions to do so. The PEP also interacts with the study team between meetings via email with any additional feedback.

## Primary outcome measures

### Clinical COVID-19 outcomes

Participants will be asked to self-report COVID-19 infection, defined as either a positive SARS-CoV-2 PCR or antibody test, or as self-reported suspected infection; the latter will be particularly relevant for those reporting historic illness early in the pandemic before widespread availability of testing. In our analyses, we will consider all those with a PCR assay for SARS-CoV-2 or a positive anti-SARS-CoV-2 serology assay as being infected. To ensure those that acquired infection prior to widespread testing availability are not excluded, in those who have never been tested by PCR or serology, we will determine infection status based on whether they, or another healthcare professional, suspected them of having had COVID-19. To investigate how the inclusion of those that report suspected (but not confirmed) COVID-19 impacts on our results, we will conduct sensitivity analyses examining only those who have undergone laboratory testing for current/previous infection.

Those reporting COVID-19 illness will be asked about: date of onset, the nature of symptoms experienced and their duration and hospitalisation (including any time spent in intensive care). Corroboration of the details of these outcomes will be possible using linked electronic healthcare records (see 'Measures obtained via data linkage', below).

### General health

This will be measured using the EQ-5D-5L instrument[23] (https://euroqol.org/), which contains five dimensions on mobility, self-care, usual activities, pain and discomfort and anxiety and depression, plus an overall self-report of health.

### Mental health

This will be measured using the Patient Health Questionnaire-2[24] for depression, the Generalised Anxiety Disorder-2[25] scale for anxiety, a three-item abbreviated version of the PCL-6 (Post Traumatic Stress Disorder Checklist-6) scale for post-traumatic stress disorder (PTSD),[26] a three-item abbreviated version of the UCLA (University of California Los Angeles) Loneliness scale[27] and an Office for National Statistics question about overall life satisfaction.[28] Participants will also be asked key questions from the Utrecht Work Engagement Scale[29] and the GMC National Training Survey questions on burnout[30] (from the Copenhagen Burnout Inventory[31]).

## Questionnaire measures

### Ethnicity

In this study, we will ask participants to self-identify the ethnic group with which they most identify using the 18

UK Census 2011 Categories.[32] The questionnaire also asks the ethnic group of any partner and of parents.

We will then collapse these 18 categories into five main ethnic categories also defined within the Census (Black, Asian, Mixed, Other, White). We will further collapse them into two groups which we will refer to as 'white' (White British, White Irish, White Gypsy or Traveller, White Other) and 'ethnic minority' (all other ethnic groups). There is currently considerable debate about the categorisation of people using ethnic groups, and in particular, the grouping of people who do not identify as white into a single category. There is also considerable debate and controversy about the words used to describe such a broad and heterogeneous grouping, with terms such as 'people of colour', 'Black Asian and Minority Ethnic' or 'BAME' used. In our choice of terms, we have followed the BMJ who in their special edition on Racism in Medicine use the term 'ethnic minority' as one that is most likely to be understood by our study population.[33] We fully acknowledge that broad ethnic groupings can mask important ethnic and cultural differences, and where possible we will use more refined ethnic groupings, while also acknowledging the heterogeneity within them.

### Nationality, religion and languages
Country of birth, nationality, parents' country of birth, grandparents' country of birth (born in UK/not born in UK), age learnt English (if second language), language(s) spoken at home (currently and as a child), religion, religiosity, ethnic identity.

### Other demographics and education
Age, gender, sex, marital status, highest level of education completed, year and country of primary professional qualification (if applicable), highest level of education achieved by parents.

### Work
Job role(s), sector(s), grade and specialty (for doctors), NHS band (for other HCWs), registered field of practice (nurses); work location(s); whether currently working, reasons for not working (if applicable); hours worked in a typical week; frequency of night shifts; contact with patients (with and without COVID-19), colleagues and others (remotely, face-to-face with social distancing, with physical contact); time spent travelling to and from work, modes of transport; access to, use of and training to use PPE; exposure to aerosol-generating procedures; NHS COVID-19 risk assessment experiences; feelings about raising a clinical concern at work, perceptions of fairness of organisation with regards career progression; redeployment as a result of the pandemic, patient exposure, training and supervision in redeployed role (if applicable); proportion of colleagues of same ethnicity to self, proportion of colleagues who are white; work engagement.

### Home and social life
Household composition (numbers, ages, relationship to participant) and sharing of accommodation; number of household members travelling using public transport or in jobs that bring them into close contact with others; childcare and support 'bubbles'; length of time at current address; type and size of accommodation including amount of shared space and access to safe outdoor space; numbers of social contacts (remotely, face to face with social distancing, with physical contact), proportion of friends of the same ethnic group to self.

### Harassment and discrimination
Experiences of discrimination in day-to-day life; discrimination at work and whether made a complaint (if applicable).

### Physical health, mental health and well-being
Height, weight, smoking and alcohol use, physical activity at work, general physical activity, change in lifestyle since start of pandemic, healthcare experience in 2019 (General Practitioner consultations and hospital admittance), influenza vaccine uptake, medication, health conditions and pregnancy, quality of life, general anxiety, depression, PTSD, loneliness, and general life satisfaction.

### COVID-19 experiences and beliefs
COVID-19 exposure, testing and test positivity; symptoms experienced, plus their severity and longevity, including diagnosis of long-COVID (if applicable); behaviour changes due to COVID-19; concern, knowledge and beliefs about COVID-19; COVID-19 information sources; enjoyment of first national lockdown (Spring 2020), COVID-19 vaccine trial participation; COVID-19 vaccination intention including offers, uptake (including vaccine brand) or reasons for refusal and vaccine beliefs.

### Trait and state psychological measures
'Big five' personality traits, locus of control, health locus of control, risk taking, burnout, personal need for structure.

### Open-ended questions
The baseline questionnaire will include three open-ended free-text questions: 'What are your thoughts on why people from ethnic minorities working in health and care have been more severely affected by COVID-19?', 'How do you see society changing as a result of COVID-19?', 'How do you see your own future changing as a result of COVID-19?'.

### Evaluation questions
Views on the length of the questionnaire and on its usefulness for understanding ethnicity and COVID-19.

### Measures obtained via data linkage
Data linkage will be used to corroborate COVID-19 clinical outcomes (acute infection, antibody positivity), major comorbidities and patterns of healthcare use.

### Biological sampling

At baseline, we will also seek consent to recontact participants in the future for DNA sampling and sampling related to immune profiling although we will submit an amendment to implement this sampling and detail the specifics relating to this at the time of submitting the amendment.

### Statistical analysis

Descriptive statistics will be calculated for the primary outcome measures and for ethnicity and key confounder/explanatory variables.

Univariate associations between ethnicity and primary outcome measures and between ethnicity and key confounders/explanatory variables calculated using $\chi^2$ tests for categorical variables, and t-tests and analyses of variance for continuous measures, with non-parametric equivalents used as appropriate for ordinal variables. This will enable the examination of the behavioural, social and clinical phenotypes of the cohort in relation to the patterns of demographics, job role, attitudes to work and work climate, social and living circumstances, values and personality and physical and mental health by ethnicity.

Using baseline data, multivariable analyses will test for associations between ethnicity and key outcomes, adjusted for the confounder/explanatory variables found to have a statistically significant univariate association with either ethnicity or the primary outcome variable(s), with interactions included as appropriate. Using follow-up data, mixed models will be used to model changes over time by ethnic group.

Models will fit ethnicity as both a binary indicator (ethnic minority vs white) and as a categorical variable based on ONS categorisation of ethnicity, with the white group used as the reference group.

## ETHICS AND DISSEMINATION
### Ethical approval

The study has been approved by the Health Research Authority (Brighton and Sussex Research Ethics Committee; ethics reference: 20/HRA/4718).

### Ethical considerations

While this study is low risk, the questionnaire covers sensitive topics around COVID-19, ethnicity (including issues of discrimination and harassment) and mental health, and these could be distressing to participants. We aim to manage this risk by clearly indicating on the PIS that the questionnaire covers sensitive topics and that participants are under no obligation to answer these, or indeed any other, questions, and provide links to support organisations.

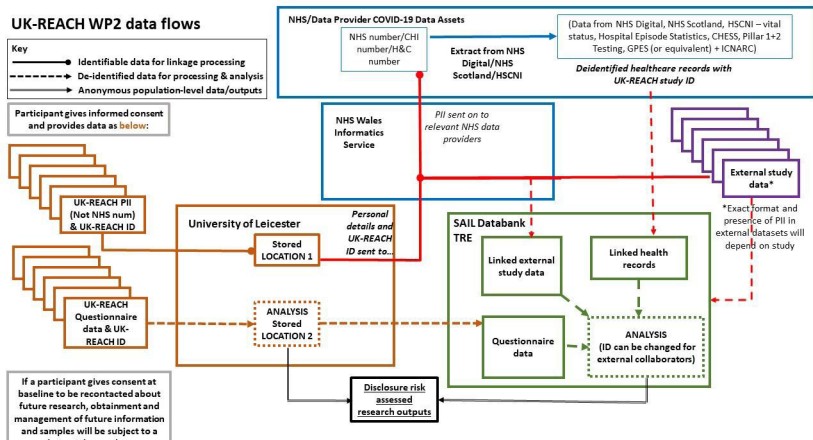

**Figure 3** Data flow diagram. Solid lines with a circle indicate identifiable data flows for linkage purposes only. Solid lines with an arrow indicate anonymous outputs. Dashed lines indicate deidentified data. After consenting to join UK-REACH, participants provide limited PII which are stored in a secure location at the University of Leicester, alongside study ID. Questionnaire data (including study ID but not alongside the aforementioned PII) are stored in a different secure location at the University of Leicester. Participants provide consent for the study to follow their health by extracting information from electronic health records. Relevant PII for each participant will be securely transferred to NHS Wales Informatics Service (alongside the UK-REACH study ID) in order to acquire NHS number/CHI number (for Scotland)/Health and Care number (for Northern Ireland). NHS/CHI/H&C number will then be used to link to healthcare records by the relevant data provider across the UK. Deidentified health records will be sent directly to the Secure Anonymised Information Linkage databank (SAIL, https://saildatabank.com), retaining Study ID but excluding PII. Questionnaire data (including study ID) will be sent from the University of Leicester to SAIL. These datasets will be linked within SAIL to provide the complete dataset. Interim analysis of unlinked questionnaire data will be performed at the University of Leicester using pseudonymised datasets. All linked data analysis will take place within SAIL. The above diagram and style was interpreted by Anna Guyatt and Chris Orton from an initial data flow diagram created and provided by Andy Boyd at the University of Bristol. It has been repurposed and amended to illustrate data flows specific to the UK-REACH project. CHESS, COVID-19 Hospitalisation in England Surveillance System; CHI, Community Health Index; GPES, General Practice Extraction Service; HSCNI, Health and Social Care Northern Ireland; ICNARC, Intensive Care National Audit & Research Centre; NHS, National Health Service; PII, personal identifiable information; UK-REACH, The United Kingdom Research study into Ethnicity And COVID-19 outcomes in Healthcare workers.

## Participant confidentiality

The participants will be identified only by a unique identifier in the main research database. Identifiable information (name, date of birth, address and so on) will be stored in a separate secure database and will be accessed only by a small number of authorised staff at the University of Leicester who require access to administer the study. All documents will be stored securely and only be accessible by study staff and authorised personnel. The study will comply with the Data Protection Act, which requires data to be anonymised as soon as it is practical to do so. Any dissemination of study findings will follow best-practice guidelines for deductive disclosure. Only aggregate data will included in publications.

## Discontinuation/withdrawal of participants from study

Participants who wish to withdraw from the cohort study will be asked to determine the desired level of withdrawal from the study as described by the two options below. We will keep a record of consent for all participants to manage recontact and for future audit. We will accept signed withdrawal forms from participants or, if they are unable to complete a withdrawal form themselves, from someone acting on the participant's behalf. At the present time, withdrawal forms will be completed electronically, but signed written forms will also be accepted when it is feasible to securely receive and store these. The options that participants will be given if they wish to withdraw:

Option 1—No further contact: we would no longer contact the participant, but would have the participant's permission to continue to obtain information by accessing their health records in the future.

Option 2—No further contact or access: we would no longer contact the participant or obtain information from the participant's health records in the future.

If participants withdraw from UK-REACH, then any data and samples already collected will remain and be used in the study. Information and data will continue to be collected about participants' health from central NHS records, hospital records and participants' GPs, unless participants state otherwise on the withdrawal form.

## Description of data flow

See figure 3 for a description of the data flow.

## Dissemination plan

Quarterly reports in months 3, 6, 9 and 12 summarising recruitment progress and initial findings on relationship between ethnicity, COVID-19 diagnosis and outcomes, physical/mental well-being and professional and social factors. Brief reports will be produced and submitted for review by the stakeholder group (see online supplemental material for details), PEP, Study Steering Committee and the UK Government's Scientific Advisory Group for Emergencies (SAGE). Papers submitted to peer reviewed journals and preprint servers.

## Author affiliations

[1]Research Department of Medical Education, University College London, London, UK
[2]Genetic Epidemiology Research Group, Department of Health Sciences, University of Leicester, Leicester, UK
[3]Department of Respiratory Sciences, University of Leicester, Leicester, UK
[4]Oxford University Hospitals NHS Foundation Trust, Oxford, UK
[5]NIHR Leicester Biomedical Research Centre Respiratory Diseases, Leicester, UK
[6]Division of Epidemiology and Public Health, University of Nottingham, Nottingham, UK
[7]University Hospitals of Leicester NHS Trust, Leicester, UK
[8]General Medical Council, London, UK
[9]Department of Health Sciences, University of Leicester, Leicester, UK
[10]Department of Infection and HIV Medicine, University Hospitals of Leicester NHS Trust, Leicester, UK
[11]Centre for Health Economics, University of York, York, UK
[12]Glenfield Hospital, NIHR Leicester Biomedical Research Centre Respiratory Diseases, Leicester, UK
[13]Diabetes Research Centre, University of Leicester, Leicester, UK

**Acknowledgements** We wish to acknowledge the PEP group and the STAG group (see online supplementary material) and SERCO as well as the following people for their support in organising the email invites from the regulatory bodies: Kerrin Clapton and Andrew Ledgard (General Medical Council), Caroline Kenny (Nursing and Midwifery Council), David Teeman and Lisa Bainbridge (General Dental Council), My Phan and John Tse (General Pharmaceutical Council), Angharad Jones (General Optical Council), Katherine Timms and Charlotte Rogers (The Health and Care Professions Council) and Mark Neale (Pharmaceutical Society of Northern Ireland).

**Collaborators** UK-REACH Study Collaborative Group: Manish Pareek (Chief investigator), Amani Al-Oraibi, Amit Gupta, Anna L Guyatt, Carl Melbourne, Catherine John, Christopher A Martin, Ian Chris McManus, Chris Orton, Claire Garwood, David Ford, Edward Dove, Fatimah Wobi, Janet Hood, Kamlesh Khunti, Katherine Woolf, Keith Abrams, Laura J Gray, Laura Nellums, Louise V Wain, Lucy Teece, Luke Bryant, Martin Tobin, Mayuri Gogoi, Osama Hassan, Robert C Free, Ruby Reed-Berendt, Sue Carr, Vishant Modhwadia.

**Contributors** MP conceived of the idea and led the application for funding with input from MDT, KK, ICM, KW, RCF, LN, SC, KRA, LJG, AG, LVW and CJ. The survey was designed by KW, MP, ICM, CM, CJ, ALG, AG, LN, CAM and RCF. Online consent and survey tools were developed by LB with support from CG and VM. KW wrote the first draft of the manuscript with input from MP and all coauthors. All authors approved the submitted manuscript.

**Funding** UK-REACH is supported by a grant from the MRC-UK Research and Innovation (MR/V027549/1) and the Department of Health and Social Care through the National Institute for Health Research (NIHR) rapid response panel to tackle COVID-19. Core funding was also provided by NIHR Biomedical Research Centres. This work is carried out with the support of BREATHE—The Health Data Research Hub for Respiratory Health (UKRI MC_PC_19004) in partnership with SAIL Databank. BREATHE is funded through the UK Research and Innovation Industrial Strategy Challenge Fund and delivered through Health Data Research UK. ALG was funded by internal fellowships at the University of Leicester from the Wellcome Trust Institutional Strategic Support Fund (204801/Z/16/Z) and the BHF Accelerator Award (AA/18/3/34220). CJ holds a Medical Research Council Clinical Research Training Fellowship (MR/P00167X/1). KRA is supported by Health Data Research (HDR) UK and as a NIHR Senior Investigator Emeritus (NF-SI-0512–10159). KK and LJG are supported by the National Institute for Health Research (NIHR) Applied Research Collaboration East Midlands (ARC EM). KK, MP and RCF are supported by the NIHR Leicester Biomedical Research Centre (BRC). CAM is an NIHR Academic Clinical Fellow (ACF-2018-11-004). LN is supported by an Academy of Medical Sciences Springboard Award (SBF005\1047). MP is supported by a NIHR Development and Skills Enhancement Award. MDT holds a Wellcome Trust Investigator Award (WT 202849/Z/16/Z) and an NIHR Senior Investigator Award. KW holds an NIHR Career Development Fellowship (CDF: CDF-2017-10-008). LVW holds a GSK/British Lung Foundation Chair in Respiratory Research (C17-1). This research was funded in whole, or in part, by the Wellcome Trust (WT204801/Z/16/Z and WT 202849/Z/16/Z).

**Disclaimer** The views expressed in the publication are those of the author(s) and not necessarily those of the National Health Service (NHS), the NIHR or the Department of Health and Social Care.

**Competing interests** SC is Deputy Medical Director of the General Medical Council, UK Honorary Professor, University of Leicester. KK is Director of the University of Leicester Centre for Black Minority Ethnic Health, Trustee of the South

Asian Health Foundation, Chair of the Ethnicity Subgroup of SAGE and Member of Independent SAGE. LVW receives grant funding from GSK and Orion, outside of the submitted work. KRA has served as a paid consultant, providing unrelated methodological and strategic advice, to the pharmaceutical and life sciences industry generally and has received unrelated research funding from Association of the British Pharmaceutical Industry, European Federation of Pharmaceutical Industries & Associations, Pfizer, Sanofi and Swiss Precision Diagnostics. He is a Partner and Director of Visible Analytics Limited, a healthcare consultancy company.

**Patient consent for publication** Not required.

**Ethics approval** Brighton and Sussex Research Ethics Committee; ethics reference: 20/HRA/4718.

**Provenance and peer review** Not commissioned; externally peer reviewed.

**ORCID iDs**
Katherine Woolf http://orcid.org/0000-0003-4915-0715
I Chris McManus http://orcid.org/0000-0003-3510-4814
Laura Nellums http://orcid.org/0000-0002-2534-6951
Christopher A Martin http://orcid.org/0000-0002-2337-4799

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
