## [Reviewer comments · BMJ Open]

ARTICLE DETAILS

TITLE (PROVISIONAL)	The United Kingdom Research study into Ethnicity And COVID-19 outcomes in Healthcare workers (UK-REACH): Protocol for a prospective longitudinal cohort study of healthcare and ancillary workers in UK healthcare settings
AUTHORS	Woolf, Katherine; Melbourne, Carl; Bryant, Luke; Guyatt, Anna; McManus, Ian; Gupta, Amit; Free, Robert; Nellums, Laura; Carr, Sue; John, Catherine; Martin, Christopher; Wain, Louise; Gray, Laura; Garwood, Claire; Modhwadia, Vishant; Abrams, Keith; Tobin, Martin; Khunti, Kamlesh; Pareek, Manish

VERSION 1 – REVIEW

REVIEWER	Rubeshkumar, Polani National Institute of Epidemiology
REVIEW RETURNED	29-Apr-2021

GENERAL COMMENTS	This is a well-structured protocol designed by a team of experts. The protocol adheres to STROBE guidelines and ethical considerations.
---

REVIEWER	Ghazy, Ramy Alexandria University High Institute of Public Health
REVIEW RETURNED	12-May-2021

GENERAL COMMENTS	Dear authors I would like to thank you for this interesting research. You intended to study the effect of ethnicity on COVID1-9 outcome. Indeed, I have few comments addressed below. Major revision 1. Regarding multilevel mode, please specify the levels you are going to analyze data based on them, i.e personal level, health facility level, district level. please keep in mind each level should consist of at least 30 units.2. Inclusion and exclusion criteria, I think signing ethical approval should not be included.3. the inclusion criteria shouldn't be repeated in exclusion. i.e inclusion of individuals aged above 16 and exclude those below 16 years.4. You mentioned that "we may adapt target sample sizes" do you mean convenience sampling method.5. You mentioned that "they will be incentivised by the inclusion of a prize" is it ethical? please confirm.6. Study duration? will it last for 8 months or 25 years?
---

	7. Study questionnaires: participants will be required to fill in 7 Questionnaires. Is it applicable? and if yes what is the duration required to do? 8. You mentioned that"s self-reported suspected infection; the latter will be particularly relevant for those reporting historic illness early in the pandemic before widespread availability of testing" Do you think this group of participants may affect the internal validity of your results? 9. Finally which tool will assess values and pesrnoality? Minor changes: 1.COVID-19, Public health < INFECTIOUS DISEASES, MENTAL HEALTH...replace < with,. 2. In the introduction, please expand SRAS-Cov-2 as it is abbreviated while it was the first time to be mentioned.
--	---

VERSION 1 – AUTHOR RESPONSE

Reviewer: 1

Dr. Polani Rubeshkumar, National Institute of Epidemiology

Response: We thank the reviewer for their comments.

Reviewer: 2

Dr. Ramy Ghazy, Alexandria University High Institute of Public Health Comments to the Author:

Response: We thank the reviewer and are pleased they agree this research is interesting.

Major revision

1. Regarding multilevel mode, please specify the levels you are going to analyze data based on them, i.e. personal level, health facility level, district level. please keep in mind each level should consist of at least 30 units.

Response: Thank you for your question which has prompted us to reflect. We have now replaced multi-level models with mixed models, which we will use to model within-participant changes over time. We have now amended this in the abstract and on p11 of the MS.

2. Inclusion and exclusion criteria, I think signing ethical approval should not be included.

Response: We are not able to include people in the study who do not consent to be included and as such, one of our inclusion criteria is "Willing and able to give informed consent."

This is standard, for example the recently published protocol by Zhang L, Shi W, Lu S, et al. (Prognostic factor analysis in patients with temporomandibular disorders after reversible treatment: study protocol for a prospective cohort study in China. *BMJ Open* 2021;11:e048011. doi:10.1136/bmjopen-2020-048011)

has as the following inclusion criterion: "Patients must volunteer to participate in the study and sign the consent form".

3. the inclusion criteria shouldn't be repeated in exclusion. i.e. inclusion of individuals aged above 16 and exclude those below 16 years.

Response: STROBE guidelines include "eligibility criteria". We have stated the inclusion and exclusion criteria for our study. We will be guided by the editors as to whether we should remove our exclusion criteria.

4. You mentioned that "we may adapt target sample sizes" do you mean convenience sampling method?

Response: Apologies that we did not make this clearer. In fact, we meant that we may adapt the sampling frame. We have now amended this on p5 of the MS.

5. You mentioned that "they will be incentivised by the inclusion of a prize" is it ethical? Please confirm.

Response: Yes, the inclusion of a prize draw was passed by the ethics committee. We have already stated that "The study has been approved by the Health Research Authority (Brighton and Sussex Research Ethics Committee; ethics reference: 20/HRA/4718)". We are uncertain whether we should explicitly state that the prize draw was included in this approval because doing so may cause readers to wonder whether other aspects of the study were not approved. We would like to be guided by the editors on this point please.

6. Study duration? will it last for 8 months or 25 years?

Response: We apologise that this is not clear. As stated in the title, our study is a prospective longitudinal cohort study. It also includes linkage to other data with consent. We state in the Abstract (p2) and Methods (p7) that we will conduct two follow-up questionnaires at 4 and 8 months, but that linkage to other data will continue for up to 25 years. We have now attempted to make this clearer by amending the paragraph in the Abstract (p2) as follows:

"A baseline questionnaire will be administered to a national cohort of UK HCWs and ancillary workers in healthcare settings, and those registered with UK healthcare regulators, with follow-up questionnaires administered at 4 and 8 months. With consent, questionnaire data will be linked to health records with up to 25 year follow-up."

We hope this is clearly explained on p7 in the section that reads:

"Participants will be invited to complete the baseline questionnaire between December 2020 and January 2021. Participants will have the option of completing two further questionnaires, one at approximately 4 months from baseline and one at approximately 8 months from baseline. Consent will be requested to follow up participants for up to 25 years (subject to funding), and during this time serial questionnaire data will be collected, and periodic linkage with healthcare records will take place (see Data linkage, below)."

7. Study questionnaires: participants will be required to fill in 7 Questionnaires. Is it applicable? And if yes what is the duration required to do?

Response: As stated on p2 and p8, we will ask participants to complete one baseline questionnaire, with additional follow-up questionnaires at 4 and 8 months. We are not sure where in our MS it is suggested that participants will complete 7 questionnaires so we are unable to clarify that part of the MS unfortunately.

8. You mentioned that"s self-reported suspected infection; the latter will be particularly relevant for those reporting historic illness early in the pandemic before widespread availability of testing" Do you think this group of participants may affect the internal validity of your results?

Response: This is an important point, which we thank the reviewer for raising. Unfortunately since widespread testing for SARS-CoV-2 was unavailable in the UK at the start of the pandemic we are unable to know whether those who thought they were infected then actually were infected (unless they subsequently had a positive antibody test, which we ask about). It is therefore possible that people who thought they were infected were not, and there were others who did not realise that they were infected. Our questionnaire is designed to try to capture all possible infections.

On p9 we have now included a sentence that reads:

"In our analyses we will consider all those with a PCR assay for SARS-CoV-2 or a positive anti-SARS-CoV-2 serology assay as being infected. To ensure those that acquired infection prior to widespread testing availability are not excluded, in those who have never been tested by PCR or serology, we will determine infection status based on whether they, or another healthcare professional, suspected them of having had COVID-19. To investigate how the inclusion of those that report suspected (but not confirmed) COVID-19 impacts upon our results, we will conduct sensitivity analyses examining only those who have undergone laboratory testing for current/previous infection."

9. Finally which tool will assess values and pesrnoality?

Response: Details of all the measures are included in the UK-REACH data dictionary (link provided on p8 of the MS). The baseline questionnaire has over 700 items, and as such we chose to only provide references in this protocol for the key variables in the study that relate directly to the main outcome measures. We would be grateful if the editors could advise whether they would like us to provide these additional references in this MS.

Minor changes:

1.COVID-19, Public health < INFECTIOUS DISEASES, MENTAL HEALTH...replace < with,.

Response: This does not appear in our MS so we assume it is automatically included when we selected key words for the study. As such, we are unable to change it.

2. In the introduction, please expand SRAS-Cov-2 as it is abbreviated while it was the first time to be mentioned.

Response: We have now done this.

Reviewer: 1

Competing interests of Reviewer: None

Reviewer: 2

Competing interests of Reviewer: No competing interest

VERSION 2 – REVIEW

REVIEWER	Ghazy, Ramy Alexandria University High Institute of Public Health
REVIEW RETURNED	21-Aug-2021
GENERAL COMMENTS	Thank you for this clarification I want to explain my second comment that consent shouldn't be one of the inclusion criteria. Consent is mandatory and I think it is better to be written under a separate section (research ethics) I recommend publication